# AutoGuide: Automated Generation and Selection of Context-Aware Guidelines for Large Language Model Agents

Yao Fu [1*]    Dong-Ki Kim [2*]    Jaekyeom Kim [2]    Sungryull Sohn [2]
Lajanugen Logeswaran [2]    Kyunghoon Bae [2]    Honglak Lee [1,2]
[1]University of Michigan    [2]LG AI Research

## Abstract

Recent advances in large language models (LLMs) have empowered AI agents capable of performing various sequential decision-making tasks. However, effectively guiding LLMs to perform well in unfamiliar domains like web navigation, where they lack sufficient knowledge, has proven to be difficult with the demonstration-based in-context learning paradigm. In this paper, we introduce a novel framework, called AUTOGUIDE, which addresses this limitation by automatically generating context-aware guidelines from offline experiences. Importantly, each context-aware guideline is expressed in concise natural language and follows a conditional structure, clearly describing the context where it is applicable. As a result, our guidelines facilitate the provision of relevant knowledge for the agent's current decision-making process, overcoming the limitations of the conventional demonstration-based learning paradigm. Our evaluation demonstrates that AUTOGUIDE significantly outperforms competitive baselines in complex benchmark domains, including real-world web navigation.

## 1 Introduction

Recent advances in large language models (LLMs) have empowered AI agents to address various sequential decision-making tasks and applications [1, 2]. The foundation of these successes involves the planning and reasoning capabilities of pre-trained LLMs, enabling agents to execute effective policies [3, 4]. The predominant approach to leveraging these (typically closed source) models for sequential decision making tasks is to provide demonstrations in the form of in-context examples. However, direct application of this learning paradigm can be limited, especially in target domains where the LLM has insufficient prior knowledge such as in web navigation, where LLM agents generally achieve low success rates due to diverse and dynamic contents [5–8]. Providing all available experiences as demonstrations to an agent can further be unsuccessful due to context length limitations, prompt sensitivity, and difficulty with complex reasoning [9–12].

On the other hand, LLMs excel in interpreting concise instructions provided as natural language, an ability that is also reinforced in the instruction-tuning phase of LLMs. Inpired by this, we explore data-driven strategies that leverage offline experiences to extract actionable knowledge to help guide LLM agents. As offline experiences implicitly convey valuable knowledge about desirable and undesirable policies in domains, they promise to serve as a useful resource for improving an LLM agent's decision-making in situations where the pre-trained LLM lacks understanding. Despite this potential benefit, a critical challenge lies in effectively extracting the implicit information embedded in offline data.

---

*Equal contribution.

38th Conference on Neural Information Processing Systems (NeurIPS 2024).

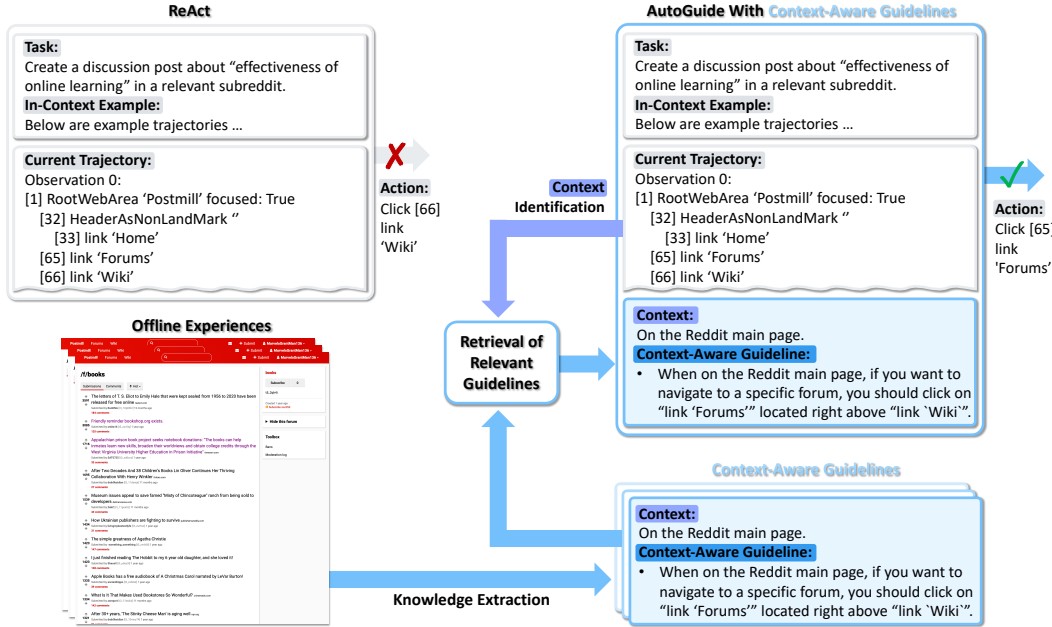

Figure 1: AutoGuide aims to extract the implicit knowledge embedded in offline experiences and help the decision-making process of an LLM agent. Specifically, our method generates a comprehensive set of context-aware guidelines from offline data and explicitly identifies when each guideline is applicable by generating its corresponding context. Our context-aware guidelines enable providing pertinent guidelines at test time by identifying the context of the current trajectory, leading to correct decision-making compared to baselines without context-aware guidelines.

To address the challenge of extracting knowledge from offline data, we propose a novel framework, called AutoGuide. Specifically, AutoGuide automatically derives a comprehensive set of context-aware guidelines from offline experiences. Our method applies these context-conditional guidelines to enhance the performance of an LLM agent by retrieving guidelines relevant to the agent's current state and incorporating them into the prompt during testing (see Figure 1). Notably, we generate context-aware guidelines in concise natural language statements, effectively compressing knowledge in offline data. Moreover, context-aware guidelines clearly describe the contexts where they are applicable, so AutoGuide enables an LLM agent to select pertinent guidelines for its current decision-making process. As a result, AutoGuide achieves the highest success rates compared to competitive baselines in complex sequential decision-making benchmark environments.

**Our contribution.** In summary, we present the following main contributions in this paper:

• **Principled method based on context-aware guidelines (Section 3):** We develop two modules to automatically generate context-aware guidelines from offline experiences: the context identification module for identifying the context of a given trajectory, and the guideline extraction module for extracting a desired guideline corresponding to that context. The outcome is a set of domain knowledge in concise natural language that enhances decision-making by providing pertinent information.

• **Comprehensive evaluation of AutoGuide (Section 4.2):** We show AutoGuide's capability in extracting helpful context-aware guidelines in various interactive benchmark domains, including navigating real-world web domains (e.g., GitHub). Our results highlight the effectiveness of AutoGuide, which significantly outperforms baselines without context-aware guidelines.

• **Analyses with important perspectives (Section 4.3):** We study various aspects of AutoGuide, such as the significance of determining the applicability of each guideline based on generated contexts. We also investigate the generalization ability of context-aware guidelines and demonstrate that our guidelines enhance the performance across out-of-domain tasks.

## 2 Related Work

**LLM-based agents.** Language models have recently been shown to possess strong priors for sequential decision-making tasks, which has given rise to LLM-powered agents [1, 2, 13, 14]. Agents

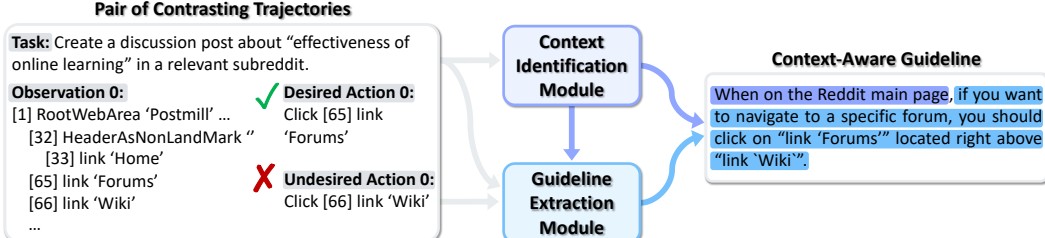

Figure 2: Context-aware guideline generation process based on a pair of contrastive trajectories $\boldsymbol{\tau}_+^i$ and $\boldsymbol{\tau}_-^i$. In this example, the two trajectories start deviating from each other at $t=0$. The context identification module generates a description of the context at $t=0$ given $\boldsymbol{\tau}_{:0}^i$, and the guideline extraction module generates the corresponding guideline for that context.

need to possess various skills to be effective in practice including planning [3, 15, 16], reasoning [4, 17], tool manipulation [18–21], code generation [22, 16], among others. In this work, we focus on building effective agents for web [23, 8] and embodied [24] environments.

**Self-reflection from past experiences.** An important capability for agents to succeed is the ability to learn from past experiences and update their behavior based on feedback. Self-feedback [25–27] has emerged as an effective technique where a model inspects its own incorrect predictions, reflects on it to identify what went wrong and attempts to improve its prediction. While self-feedback provides intra-task (i.e., per-episode) knowledge within a task based on immediate feedback, our approach offers an orthogonal and complementary aspect of inter-task knowledge (over multiple tasks) by considering multiple train tasks in offline data. AutoGuide enhances learning efficiency and credit assignment by utilizing detailed feedback from multiple tasks. However, these self-feedback approaches are complementary to AUTOGUIDE and can be used in conjunction with our approach, as shown in our experiments (see Section 4.2).

**Leveraging natural language guidance.** Natural Language can be a rich source of information for agents to learn to act efficiently. Prior work has explored the notion of learning from human-written text manuals, which describe details about the environment [28–30]. Recent work has explored automatically generating such guidance in the form of chain-of-thought reasoning [4, 23], which emulates a thought process or rationale for agent's predictions. In contrast to approaches which generate such guidance dynamically on the fly by imitating example guidance demonstrations provided by a human, our approach carefully compares trajectories in offline data to generate appropriate guidance and uses these guidelines for predicting better actions. ExpeL [31] proposed a related approach to derive guidelines. In contrast to ExpeL, where all guidelines are provided to an agent as a prompt, our guideline selection process is contextual, where guidelines relevant to the agent's current state are retrieved and used for prediction. We show that this substantially improves over ExpeL's non-contextual guideline-based approach.

## 3  AUTOGUIDE: Principled Method Based on Context-Aware Guidelines

Our work is motivated by the increasing availability of offline experiences that agents or humans naturally accumulate through their interactions with the environment. AUTOGUIDE aims to leverage this offline data to improve the decision-making of an LLM agent by generating helpful context-aware guidelines. This section details how AUTOGUIDE automatically constructs these guidelines and applies them to guide action generation at test time.

### 3.1  Problem Statement

Formally, AUTOGUIDE is given offline data $\mathcal{D}_{\text{train}} = (\boldsymbol{\tau}^1, ..., \boldsymbol{\tau}^N)$ that consist of $N$ trajectories from training tasks. Each trajectory $\boldsymbol{\tau} = (x_0, a_0, r_0, ..., r_T)$ is a sequence of observations, actions, and rewards following the partially observable Markov decision process [32]. The return of a trajectory is defined as the sum of rewards obtained throughout the trajectory: $R(\tau) = \sum_{t=0}^{T} r_t$. The objective of AUTOGUIDE is to distill knowledge from offline experiences into a useful natural language format, such that the extracted information helps to maximize the expected return $\mathbb{E}_\tau[R(\tau)]$ during test time.

**Algorithm 1** Extracting context-aware guidelines from offline data

**Input:** Offline data $\mathcal{D}_{\text{train}}$, context identification module $\mathcal{M}_{\text{context}}$, guideline extraction module $\mathcal{M}_{\text{guideline}}$
Initialize context-aware guideline dictionary $\mathcal{G}$
**for** Each pair $\boldsymbol{\tau}_+^i, \boldsymbol{\tau}_-^i \in \mathcal{D}_{\text{train}}$ **do**
  # Identify the context from a trajectory
  Find the deviating timestep $t$ from $\boldsymbol{\tau}_+^i$ and $\boldsymbol{\tau}_-^i$
  CONTEXT $\leftarrow \mathcal{M}_{\text{context}}(\boldsymbol{\tau}_{:t}^i)$
  # Check if the current context matches any existing contexts
  **if** CONTEXT $\notin \mathcal{G}$ **then**
    $\mathcal{G}[\text{CONTEXT}] = \{\}$
  **end if**
  # Generate the context-aware guideline
  GUIDELINE$\leftarrow \mathcal{M}_{\text{guideline}}(\boldsymbol{\tau}_+^i, \boldsymbol{\tau}_-^i, \text{CONTEXT})$
  $\mathcal{G}[\text{CONTEXT}] \leftarrow \mathcal{G}[\text{CONTEXT}] \cup \{\text{GUIDELINE}\}$
**end for**
**Return** Context-aware guideline dictionary $\mathcal{G}$

**Algorithm 2** Applying context-aware guidelines at test time

**Input:** Context-aware guideline dictionary $\mathcal{G}$, context identification module $\mathcal{M}_{\text{context}}$, guideline selection module $\mathcal{M}_{\text{select}}$, LLM agent policy $\pi$
Initialize test trajectory $\boldsymbol{\tau} = \{x_0\}$
**for** Each timestep $t$ **do**
  # Identify the current context from a trajectory
  CONTEXT $\leftarrow \mathcal{M}_{\text{context}}(\boldsymbol{\tau})$
  # If the current context matches any existing ones, perform top-$k$ guideline selection
  **if** CONTEXT $\in \mathcal{G}$ **then**
    GUIDELINES $\leftarrow \mathcal{M}_{\text{select}}(\text{CONTEXT}, \boldsymbol{\tau}; \mathcal{G}, k)$
  **else**
    GUIDELINES $\leftarrow \varnothing$
  **end if**
  # Action selection based on guidelines
  $a_t \sim \pi(\boldsymbol{\tau}, \text{CONTEXT}, \text{GUIDELINES})$
  Execute action $a_t$ and observe $x_{t+1}$
  Update trajectory $\boldsymbol{\tau} \leftarrow \boldsymbol{\tau} \cup \{\text{CONTEXT}, a_t, x_{t+1}\}$
**end for**

## 3.2 Extraction of Context-Aware Guidelines

AUTOGUIDE generates a set of context-aware guidelines by utilizing pairs of contrastive trajectories from offline data. Each context-aware guideline is expressed in concise natural language and follows a conditional structure, clearly describing the context in which the guideline is applicable. Intuitively, contrasting a pair of trajectories with different returns provides important information about when and which actions are effective or ineffective in maximizing expected returns. Building on this insight, we develop two modules for automatically extracting context-aware guidelines (see Figure 2):

**Context identification module.** This module is responsible for abstracting the given partial trajectory into its *context*, a concise natural language description of the agent's state. More specifically, for a timestep $t$ and the corresponding trajectory $\boldsymbol{\tau}_{:t}^i := (x_0, a_0, ..., x_t)$, we prompt LLMs to clearly describe the agent's status:

$$\text{CONTEXT} \leftarrow \mathcal{M}_{\text{context}}(\boldsymbol{\tau}_{:t}^i), \tag{1}$$

Our prompt templates for the context identification module are shown in Appendix C.1.

**Guideline extraction module.** This module aims to generate a desired guideline corresponding to a specific context. Let $\boldsymbol{\tau}_+^i$ and $\boldsymbol{\tau}_-^i$ represent a contrasting pair of trajectories for the same task $i$ in offline data $\mathcal{D}_{\text{train}}$, where $R(\boldsymbol{\tau}_+^i) > R(\boldsymbol{\tau}_-^i)$. We want to contrast the pair of trajectories to find desired behaviors at an important timestep. To do this, we compare these two trajectories to find the deviation timestep $t$ at which they begin to diverge due to different actions. Then we apply the context identification module to summarize the context for the shared part of the trajectory $\boldsymbol{\tau}_{:t}^i$. Eventually, we extract a useful natural language guideline by examining the paired contrastive trajectories $\boldsymbol{\tau}_+^i$ and $\boldsymbol{\tau}_-^i$ with respect to the context:

$$\text{GUIDELINE} \leftarrow \mathcal{M}_{\text{guideline}}(\boldsymbol{\tau}_+^i, \boldsymbol{\tau}_-^i, \text{CONTEXT}), \tag{2}$$

where we refer to Appendix C.2 for our prompt template. As an example, the paired trajectories in Figure 2 deviate from timestep $t = 0$, for which the context is summarized as *On the Reddit main page*. This module then generates the following context-aware guideline: *When on the Reddit main page, if you want to navigate to a specific forum, you should click on "link 'Forums'" located right above "link 'Wiki'"*.

**Construction of context-aware guidelines.** We collect context-aware guidelines $\mathcal{G}$ by iterating through available pairs in the paired offline data and organize the guidelines in a dictionary format, using the context as the key and the corresponding guidelines as the value (see Algorithm 1). In particular, we observe that the context identification module occasionally produces contexts that describe the same situation but are expressed slightly differently. To minimize redundancy, we employ an LLM to determine if the current context corresponds to any previously identified context.

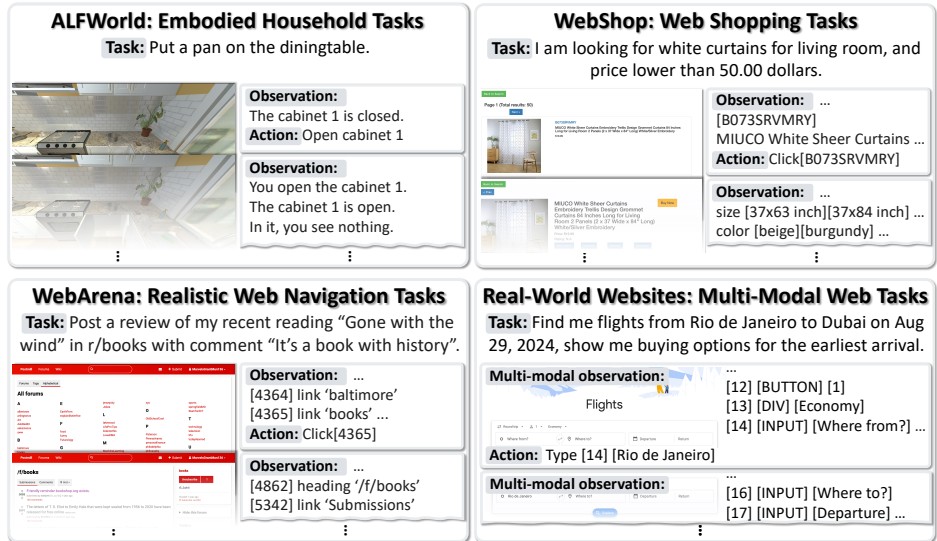

Figure 3: Sequential decision-making benchmark domains considered in our work: ALFWorld [24], WebShop [33], WebArena [8], and multi-modal real-world websites. Graphic credit: [34, 33, 8, 35].

If a match is found, we reuse the existing context; otherwise, we introduce a new context into our dictionary $\mathcal{G}$. The specific prompt template for this context-matching procedure is in Appendix C.3.

### 3.3 Applying Context-Aware Guidelines at Test Time

After extracting a set of context-aware guidelines $\mathcal{G}$ from offline experiences, our method employs these guidelines to enhance the decision-making of an LLM agent during testing. At each timestep, AUTOGUIDE identifies the CONTEXT of the current test trajectory $\tau$ up to timestep $t$ (which represents the agent's interactions up to the current time-step) using our context identification module $\mathcal{M}_{\text{context}}$. Our guideline selection module $\mathcal{M}_{\text{select}}$ then selects relevant guidelines for CONTEXT from $\mathcal{G}$. More specifically, the module applies CONTEXT as the key to fetch a set of possible guidelines $\mathcal{G}[\text{CONTEXT}]$. If there are more than $k$ guidelines in $\mathcal{G}[\text{CONTEXT}]$, $\mathcal{M}_{\text{select}}$ prompts an LLM to choose top-$k$ guidelines for the specific $\tau$:

$$\text{RELEVANT GUIDELINES} \leftarrow \mathcal{M}_{\text{select}}(\text{CONTEXT}, \tau; \mathcal{G}, k), \tag{3}$$

where Appendix C.4 details the prompt template for this selection procedure. Subsequently, AUTOGUIDE incorporates both the context and relevant guidelines into the agent's action generation prompt. Therefore, the agent selects an action by considering the provided context and guidelines (see Figure 1 for an example). This process iterates until the end of the test trajectory (see Algorithm 2).

**Key benefits of AUTOGUIDE.** First, the extraction of context-aware guidelines in AUTOGUIDE offers the inherent benefit of providing relevant guidelines for the context of interest. This capability is important since neglecting the specific context in which a guideline applies can confuse the agent's decision-making process. The second key benefit is the generation of concise natural language guidelines, which can be seamlessly incorporated into any prompt-based LLM agent. Lastly, AUTOGUIDE generates guidelines at the individual context level rather than at the trajectory level. Given that a single incorrect action can lead to a complete failure, it is essential to provide detailed assistance in each action selection process. With these advantages, we demonstrate in the next section that our approach significantly enhances the performance of LLM agents.

## 4 Evaluation

This section demonstrates the efficacy of AUTOGUIDE by conducting experiments on a diverse suite of sequential decision-making benchmark domains. We also perform important analyses about AUTOGUIDE, such as the ablation study of different AUTOGUIDE components, comparison to

in-context learning, and generalization to out-of-domain tasks. We refer to Appendix B for additional experimental details.

## 4.1 Evaluation Setup

### 4.1.1 Sequential Decision-Making Benchmark Domains

We consider the following interactive sequential decision-making benchmarks to study various aspects of AUTOGUIDE (see Figure 3):

• **ALFWorld [24]:** In this embodied benchmark, an LLM agent interacts with an environment to carry out household tasks, such as placing a pan on the dining table. Observations and actions are expressed in natural language statements, and the agent must navigate through the space and manipulate objects to successfully complete the tasks.

• **WebShop [33]:** This interactive web environment simulates the task of online shopping on an e-commerce website. The agent's goal is to understand a text instruction and buy a product that meets specified criteria. This involves querying the website's search engine, understanding the descriptions and details of each item, and selecting necessary options.

• **WebArena [8]:** This web-based benchmark introduces realistic environments by replicating the functionality and data found in popular web domains (e.g., Gitlab, Reddit, Wikipedia). Compared to WebShop, WebArena presents more challenges and difficulties for an LLM agent due to its large observation and action space, along with tasks that involve longer planning horizons. We focus oe the Reddit domain for the main WebArena experiments.

• **Real-world multi-modal websites:** Finally, we consider evaluating AUTOGUIDE on a variety of real-world website tasks. These span from a collaborative software development platform (e.g., GitHub) to a flight search engine (e.g., Google Flights) and an online education platform (e.g., Coursera). Please refer to Appendix B.4 for example tasks. In particular, in comparison to WebShop and WebArena, we design our tasks to be multi-modal such that the agent must consider both visual (e.g., images) and textual information (e.g., HTML) to complete these tasks.

### 4.1.2 Baselines

We compare AUTOGUIDE against the following baseline approaches to study the effect of context-aware guidelines (refer to Appendix B for more details):

• **ReAct [23]:** This LLM-based planning method integrates reasoning and acting to address sequential decision-making tasks. However, it does not leverage offline experiences and thus suffers from the limited understanding of pre-trained LLMs in downstream domains.

• **ExpeL [31]:** This method also extracts natural language knowledge from offline data. However, it fails to consider the applicability of guidelines and does not generate context-aware guidelines. Instead, it provides all guidelines to an LLM agent without filtering out irrelevant ones based on the current context. ExpeL has two contributions, the guideline generation and in-context example selection module. Because the latter is orthogonal to our analysis and can be seamlessly combined with our method, we consider ExpeL with guidelines in our experiments.

• **Reflexion [27]:** This approach converts environmental feedback into text statements to assist an LLM agent (e.g., ReAct) in the next trial. The baseline generates valuable feedback about solving a specific test task. We demonstrate how context-aware guidelines derived by AUTOGUIDE can be combined with the feedback.

### 4.1.3 Implementation

We collect offline experiences either by running ReAct and Reflexion, or incorporating human demonstrations. We use ReAct with GPT-3.5-turbo as our base LLM agent for WebShop and ALFWorld and GPT-4-turbo for WebArena. For each benchmark, we apply the same GPT model for action generation, context identification, and guideline selection. We extract context-aware guidelines from offline data with GPT-4-turbo and evaluate their effectiveness by applying them to the test set with non-overlapping tasks. We refer to Appendix B for more details.

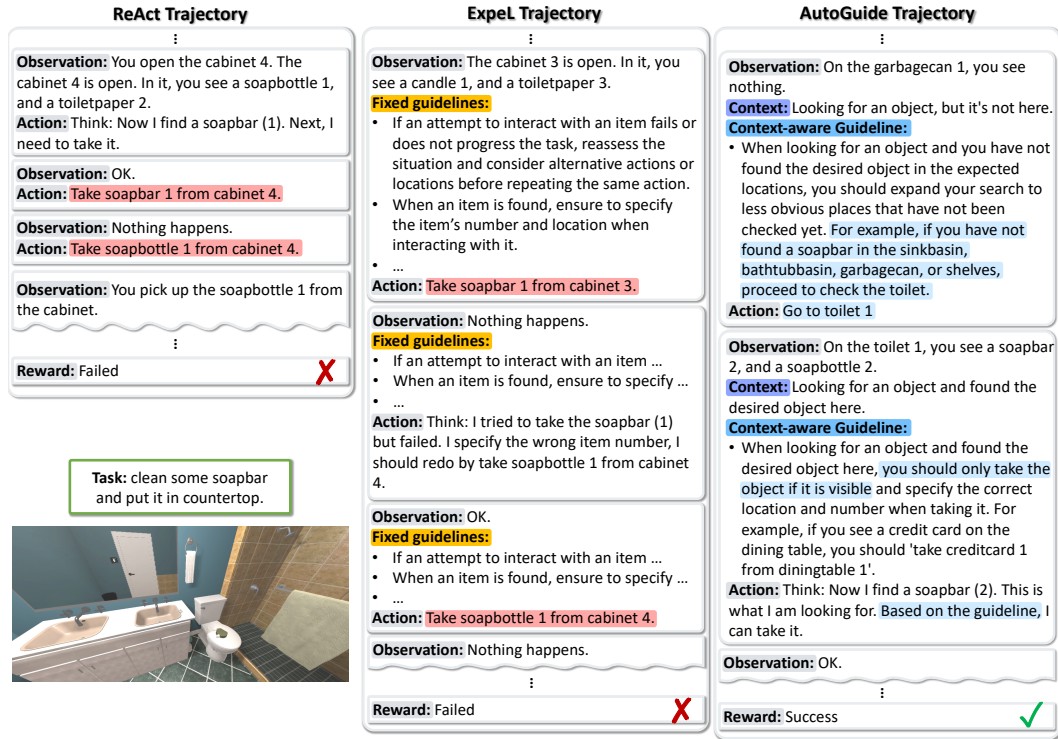

Figure 4: Trajectories of ReAct, ExpeL, and AUTOGUIDE from the same test task. ReAct (Left) chose the wrong item, consequently failing the task in the end. ExpeL (middle) was confused by guidelines that were irrelevant to current context, leading to incorrect reasoning and actions. AUTOGUIDE (right) selects relevant guidelines to the agent's context, enabling the agent to accomplish the task.

| Algorithm | | Offline data? | Context aware? | ALFWorld [24] | WebShop [33] | | WebArena [8] |
|---|---|---|---|---|---|---|---|
| | | | | Success Rate (SR)↑ | Reward↑ | SR↑ | SR↑ |
| ReAct [23] | | ✗ | ✗ | 54.5% | 66.4 | 30% | 8.0% |
| ExpeL [31] | | ✓ | ✗ | 59.0% | 60.9 | 35% | 21.8% |
| AUTOGUIDE | | ✓ | ✓ | **79.1%** | **73.4** | **46%** | **47.1%** |
| ReAct [23] | + Reflexion [27] | ✗ | ✗ | 67.2% | 77.1 | 51% | N/A |
| ExpeL [31] | + Reflexion [27] | ✓ | ✗ | 71.6% | 71.7 | 42% | N/A |
| AUTOGUIDE | + Reflexion [27] | ✓ | ✓ | **88.1%** | **81.4** | **57%** | N/A |

Table 1: Test reward and success rate on ALFWorld, WebShop, and WebArena. The base agent model for ALFWorld and WebShop is GPT-3.5-turbo and for WebArena is GPT-4-turbo. Reflexion is done by GPT-4-turbo for at most 3 trials. In our experiments, due to token limit of GPT, we did not experiment with Reflexion on WebArena tasks.

## 4.2 Main Results

**Q1.** *How effective is* AUTOGUIDE *compared to baselines without context-aware guidelines?*

To answer this question, we compare methods on ALFWorld, WebShop, and WebArena benchmarks. The performance on the test datasets is presented in Table 1. There are three notable observations:

1. **Effectiveness of context-aware guidelines.** Our approach surpasses baseline performance in both ALFWorld and WebShop, achieving the highest test rewards and success rates in Table 1. These results highlight the effectiveness of employing context-aware guidelines in language-based decision-making domains. To further examine the action selection differences among ReAct, ExpeL, and our method, we present their trajectories in Figure 4. We observed that ReAct makes common mistakes such as trying to take soapbar that is not visible, or taking a soapbottle instead of soapbar due to their similar names. Both ExpeL and AUTOGUIDE improve on this by extracting guidelines from similar mistakes in the offline experience. However, ExpeL often erroneously applies incorrect guidelines

| Algorithm | GitHub | Flights | Coursera |
|---|---|---|---|
| | SR↑ | SR↑ | SR↑ |
| SoM [36] | 2/30 | 5/20 | 1/20 |
| AUTOGUIDE | **19/30** | **9/20** | **14/20** |

Table 2: Test results of AUTOGUIDE on 3 real-world web domains within multi-modal settings. The base agent model runs with GPT-4V and applying context-aware guidelines significantly improves the performance.

| Algorithm | WebShop | |
|---|---|---|
| | Reward↑ | SR↑ |
| ReAct (1-shot) | 66.4 | 30% |
| ReAct (2-shot) | 66.0 | 35% |
| ReAct (4-shot) | 70.2 | 37% |
| ReAct (6-shot) | 71.0 | 38% |
| AUTOGUIDE | **73.4** | **46%** |

Table 3: Analysis of AUTOGUIDE against ReAct with varying numbers of in-context examples.

| Top-$k$ | WebShop |
|---|---|
| | SR↑ |
| $k=0$ | 30% |
| $k=1$ | 42% |
| $k=2$ | 46% |
| $k=3$ | 47% |
| $k=5$ | 43% |

Table 4: Ablation study of AUTOGUIDE using various top-$k$ values.

due to the availability of all guidelines at each timestep. In Figure 4, ExpeL mistakenly attends to the second guideline *"ensure to specify the item's number and location ..."*, leading to wrong reasoning and action. AUTOGUIDE presents relevant guidelines at necessary moments, enabling accurate task completion by avoiding the mistakes seen in ExpeL and ReAct.

2. **Importance of providing pertinent knowledge.** ExpeL approach helps ReAct by extracting knowledge from offline experiences, but its impact is not as significant as AUTOGUIDE. Recall that for ExpeL, the guidelines are neither generated for specific contexts at training time nor selected to only provide context-aware guidelines at test time. As a result, irrelevant guidelines can be introduced to an agent, potentially causing confusion for the agent. Consequently, the result highlights the significance of providing relevant guidelines conditioned on contexts for LLM agents.

3. **Scalability to complex environments.** We conduct experiments on WebArena-Reddit, which features more diverse tasks on realistic and complex websites requiring longer action sequences. This domain has a larger observation space and a more complex action space (e.g., scrolling). Table 1 presents the results, where AUTOGUIDE achieves the highest success rate with a significant margin when compared to ReAct and ExpeL. We observe that ReAct scores low task success rate (8.0%) in WebArena due to the complex observation and action spaces and longer task horizon. In ExpeL, the issue of presenting all guidelines to an agent is exacerbated in the WebArena compared to simpler environments like ALFWorld and WebShop. WebArena's wide variety of tasks across different domains requires a larger number of guidelines to cover the knowledge needed for all tasks and domains. This results in either an overload of irrelevant guidelines that could mislead the agent or a lack of crucial information when the number of guidelines is limited, as suggested in ExpeL [31]. In contrast, AUTOGUIDE achieves a more significant performance enhancement (47.1%) compared to ExpeL (21.8%) by efficiently providing pertinent guidelines and minimizing the burden on context capacity. We refer to Figure 14 for a list of example contexts and guidelines.

**Q2.** *How does AUTOGUIDE perform when combined with test-time self-feedback approaches?*

Our context-aware guidelines effectively provide *inter-task* knowledge by considering multiple tasks in offline data. Meanwhile, self-feedback methods (e.g., Reflexion) offer *intra-task* knowledge based on environmental feedback during test time. In this question, we explore the effectiveness of integrating both inter-task and intra-task information. The results presented in Table 1 demonstrate that the combination of AUTOGUIDE with Reflexion achieves the highest performance in the WebShop and ALFWorld benchmarks. Hence, we find that our context-aware guidelines positively complement the intra-task knowledge of Reflexion. Another observation from Table 1 is that, while ExpeL + Reflexion outperforms ExpeL alone, this combination is not as effective as other approaches. This limitation may stem from ExpeL introducing irrelevant knowledge, potentially leading to conflicts with Reflexion's feedback and having an adverse impact on the decision-making process.

**Q3.** *Can AUTOGUIDE generate context-aware guidelines for multi-modal inputs?*

Going beyond text-only inputs is an essential step toward building capable agents for solving real-world environments and tasks. We test AUTOGUIDE in a complex multi-modal setting, where each observation includes image and text information. Specifically, we introduce a set of real-world website navigation tasks in 3 domains: GitHub, Google Flights, and Coursera. For these multi-modal tasks, we employ the Set-of-Marks (SoM) agent [36, 5] as our base method. The SoM prompting improves the visual grounding capabilities of large multi-modal models such as GPT-4V by adding visually distinguishable marks to image inputs [36]. We apply AUTOGUIDE with GPT-4V to generate natural language context-aware guidelines from collected trajectories with both image and text observations.

| Algorithm | WebArena–Shopping |
| --- | --- |
| | SR↑ |
| ReAct | 10.2% |
| AUTOGUIDE | 20.4% |

| Algorithm | CI | GES | WebShop SR↑ |
| --- | --- | --- | --- |
| ReAct | ✗ | ✗ | 30% |
| ReAct + CI | ✓ | ✗ | 36% |
| ReAct + GES | ✗ | ✓ | 37% |
| AUTOGUIDE | ✓ | ✓ | **46%** |

Table 5: Out-of-distribution generalization of context-aware guidelines from WebShop on the 98 WebArena–Shopping tasks that have a product in the intent template.

Table 6: Ablation study of AUTOGUIDE, analyzing each module's contribution in WebShop. CI denotes our context identification module, and GES denotes the guideline extraction and selection modules.

Table 2 shows the effectiveness of AUTOGUIDE, demonstrating its generalization ability to complex real-world multi-modal settings. We refer to Figure 15 for example context-aware guidelines.

### 4.3 Analyses of AUTOGUIDE

**Q4.** *How does* AUTOGUIDE *compare to ReAct with varying numbers of in-context examples?*

Table 3 shows that, while increasing the number of in-context examples for ReAct gradually improves performance, there is a plateau at a certain number of shots. Additionally, ReAct with more than 6 shots often exceeds the token limit of GPT-3.5-turbo. These results indicate that directly inputting raw trajectories into ReAct for in-context learning is not an effective way to fully leverage offline data. In contrast, AUTOGUIDE extracts knowledge from entire training trajectories by summarizing them into concise context-aware guidelines, making them easy to integrate with prompt-based agents.

**Q5.** *How does altering the number of top-$k$ guidelines impact the performance of* AUTOGUIDE?

We conducted an ablation study on WebShop using various values of $k$ in Table 4. We find that employing context-aware guidelines consistently outperforms the no-guideline baseline ($k = 0$; ReAct). The $k = 3$ yields the best performance. The largest $k$ value of 5 can lead an LLM agent to overthink, potentially resulting in a slight decrease in performance. Conversely, a smaller $k$, like $k = 1$, may cause LLM to overlook additional helpful guidelines, leading to slightly worse performance.

**Q6.** *How do* AUTOGUIDE*'s context-aware guidelines generalize to out-of-domain environments?*

We conduct an experiment to further demonstrate AUTOGUIDE's out-of-domain capability across different domains but relevant tasks. We extract context-aware guidelines from WebShop and apply them to WebArena-Shopping, which is a distinct domain with variations in observation/action spaces, task intentions, and episodic horizons. For this domain adaptation case, we additionally incorporate a grounding module to align the context-aware guidelines from WebShop to WebArena's observations based on GPT-4-Turbo. As shown in Table 5, the transferred guidelines bring a notable improvement in success rates compared to the ReAct baseline in WebArena Shopping.

**Q7.** *How does each component of* AUTOGUIDE *contribute to the final results?*

We evaluate the impact of different components within AUTOGUIDE on its performance in WebShop, as detailed in Table 6. We examine two variants: ReAct+CI and ReAct+GES. The ReAct+CI, which incorporates contexts into observations without guidelines, shows improvement over ReAct. This suggests that contexts enhance decision-making by verifying the current state before action selection. ReAct+GES, which generates guidelines from trajectories without contexts and employs GPT-3.5-turbo for guideline selection, also enhances performance but is less effective than the full AUTOGUIDE. This indicates that choosing relevant guidelines based on the trajectory alone is more challenging than using contexts. Therefore, integrating both context summaries and guidelines is crucial for maximizing the benefits of AUTOGUIDE.

## 5   Conclusion

We present AUTOGUIDE, an effective framework for exploiting important domain knowledge from offline experiences for improving decision-making with pre-trained LLMs. We proposed to generate context-aware guidelines that can be incorporated into prompts for LLM agents. As AUTOGUIDE extracts the guidelines by contrasting trajectories in offline data, the resulting context-aware guidelines carry critical information for preventing failures in the domains. For inference, it provides the guidelines pertinent to each of the different context that LLM agents encounter, which can make

pre-trained LLMs strong decision-making agents in the downstream domains. Empirically, we showed that AUTOGUIDE outperforms strong baselines by a large margin and achieves outstanding performance in decision-making benchmarks.

## 6 Acknowledgements

This work was supported in part by LG AI Research.

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

# A    Limitation and Broader Impacts

**Limitation.** The performance of AUTOGUIDE depends on the diversity of offline experiences. As such, one important direction for improvement is to automatically collect diverse offline experiences through continual learning, where we iteratively generate guidelines and use them to gather more trajectories with high rewards. Another avenue is the need for quantifying the quality of generated contexts and guidelines. Currently, apart from applying context-aware guidelines to ReAct and measuring test time performance, there lacks a standardized method for quantifying the quality of generated contexts and selected guidelines. Introducing a quantifiable metric to approximate the quality could pave the way for new optimization approaches such as reinforcement learning.

**Broader impact.** This paper introduces research aimed at enhancing the decision-making capabilities of LLMs. In terms of societal impact, while we develop a generic LLM-based autonomous agent, having biased offline datasets may lead to making decisions with suboptimal outcomes. Additionally, autonomous agents may be misused for malicious applications. To mitigate these risks, potential solutions would include diversifying datasets, implementing ethical oversight, ensuring transparency and accountability, engaging with stakeholders for a broader perspective, and incorporating security measures to prevent misuse. We believe that the research community, including ourselves, should responsibly advance LLM-based agent research, prioritizing societal well-being and ethical considerations.

# B    Evaluation Details

## B.1    ALFWorld [24]

### B.1.1    Environment details

Each task in ALFWorld starts with a description of the specific environment and the goal to achieve. At each timestep, an agent can choose one of the following actions to interact with the objects and receptacles in the environment:

- go to [recep]
- take [object] from [recep]
- put [object] in/on [recep]
- open/close/use [recep]
- clean/heat/cool [object] with [recep]

Alternatively, the agent can generate a think action for planning and reflection, which helps with decision-making but does not change the environment itself. After one action is performed, the environment returns an observation that describes view changes.

Following ReAct, we concatenate a list of (observation, action) pairs to show the entire trajectory up to the current timestep for LLM agents to generate the next action. We experiment on 134 unseen test tasks with 6 categories of pick_and_place, pick_clean_then_place, pick_heat_then_place, pick_cool_then_place, look_at_obj, and pick_two_obj. For each task, the agent is allowed to take a maximum of 50 actions.

### B.1.2    Baseline and Models

For ALFWorld tasks, we follow the same setting as ReAct by providing 2 in-context examples for each of the 6 task categories. The original results of ReAct in their paper are produced based on Text-Davinci-002. However, this GPT version is no longer available, so we apply gpt-3.5-turbo-instruct instead to generate actions. For ExpeL, we directly take the guidelines from their appendix and append them to the ReAct agent at test time.

### B.1.3    Implementation details of AUTOGUIDE

We run the first 100 training tasks of ALFWorld to collect $(\tau_+, \tau_-)$ pairs with ReAct+Reflexion and extract context-dependant guidelines on the collected data. For context identification, we provide

2-shot demonstrations for each of the 6 task categories. The corresponding prompt templates can be found in appendix C. All parameter details are shown in table 7.

| Parameter name | Value |
|---|---|
| Allowed Episode Length | 50 |
| n-shots | 2 |
| Agent Model | gpt-3.5-turbo-instruct |
| Context Identification Model | gpt-3.5-turbo-instruct |
| Guideline Selection Model | gpt-3.5-turbo-instruct |
| Guideline Extraction Model | gpt-4-1106-preview |
| Reflexion Model | gpt-4-1106-preview |
| top-k guideline selection | 2 |

Table 7: Experiment hyperparameters on ALFWorld. The maximum allowed episode length and n-shots follow the same setup in ReAct.

## B.2 WebShop [33]

### B.2.1 Environment details

WebShop provides an e-commerce environment, where the objective is to find and buy the product that matches the task-specific Instruction. The agent can select one of the following actions to perform:

- search[query]
- click[button]

Following ReAct, the agent can generate think actions to do planning or reflection. After buying a product, the environment returns a reward showing how well the bought product matches the target one in type, price, buying options, and attributes. The reward is calculated by:

$$r = r_{type} \cdot \frac{|U_{att} \cap Y_{att}| + |U_{opt} \cap Y_{opt}| + \mathbb{1}[y_{price} \leq u_{price}]}{|U_{att}| + |U_{opt}| + 1}$$

where y is the bought product and u is the target product. Same as ALFWorld, for WebShop, the agent takes $(\text{obs}_t, \text{act}_t)$ pairs for every previous timestep $t$ as input to generate the next action.

### B.2.2 Baseline and Models

Following ReAct, experiments are done in a one-shot setting. We apply gpt-3.5-turbo-0613 to generate actions, but when the token number exceeds the token limit (for example, for the n-shot ReAct experiments in table 1), we use the 16k version of gpt-3.5-turbo-0613 instead. For ExpeL, we could not find how many training tasks the framework used for training. Therefore, we directly apply the guidelines from the appendix of their paper at test time. We only consider ExpeL with guidelines, not ExpeL with in-context example selection in our experiments for a fair comparison. The in-context example selection method is orthogonal to our work and can be easily combined with our method. For Reflexion, as shown in their paper, their 2-shot Reflexion prompt does not work well on WebShop. Therefore, we re-write the prompt and apply gpt-4-1106-preview to generate episode-level reflections for all Reflexion experiments. Following Reflexion and ExpeL, the evaluation is done on 100 test tasks. The maximum number of allowed actions for each task is 15. At the same time, each search action shows the top 3 products for the search query. Please refer to Table 8 for more details about the experiments.

### B.2.3 Implementation details of AUTOGUIDE

We randomly sample 50 training tasks from the training set of WebShop, on which we run ReAct+Reflexion to collect pairs and generate guidelines. The context identification prompt is one-shot, which is shown in Appendix C. At test time, we ask gpt-3.5-turbo-0613 to select each state's most relevant top 2 guidelines.

| Parameter name | Value |
| --- | --- |
| Allowed Episode Length | 15 |
| # of Search Results | 3 |
| n-shots | 1 |
| Agent Model | gpt-3.5-turbo-0613 |
| Context Identification Model | gpt-3.5-turbo-0613 |
| Guideline Selection Model | gpt-3.5-turbo-0613 |
| Guideline Extraction Model | gpt-4-1106-preview |
| Reflexion Model | gpt-4-1106-preview |
| top-k guideline selection | 2 |

Table 8: Experiment hyperparameters on WebShop. The maximum allowed episode length, the number of search results per page, and n-shots follow the same setup in ReAct.

## B.3    WebArena [8]

### B.3.1    Environment details

WebArena provides web-based benchmark environments that closely follow the data and functionality of real-world websites. Unlike other benchmarks like WebShop that provide clean text of website information as observation, WebArena's webpage content is represented as an accessibility tree, which is a subset of the DOM tree with useful elements of a webpage. For our expeirments, we focus on WebArena Reddit, which simulates the real Reddit websites with users, forums, and posts with abundant text information.

For each task in WebArena, the agent is expected to achieve a task-specific intent. At each timestep, WebArena provides a list of opened tabs, the accessibility tree of the focused webpage, and the URL of the current page as observation. For WebArena, each observation is long. Therefore, following the baseline in WebArena, at each timestep, we only provide the observation of the current timestep to the agent. We additionally provide up to 5 past actions for the agent to understand what it did in the past. The allowed actions in WebArena include the following:

- goto [url]
- click [element_id]
- type [element_id] [text] [1 for enter or 0 for not enter]
- press [key_combination]
- scroll [up or down]
- go_back

The maximum allowed number of actions for a single task is 20. Note that WebArena does not provide training tasks, but the work provides 19 demonstrations for Reddit, each of a different category. Therefore, we set these 19 tasks as the training tasks and then test on the rest 87 tasks.

### B.3.2    Baseline and Models

We directly run the two-shot ReAct-style baseline in the official codebase of WebArena using gpt4-preview-1106. For ExpeL, the original paper does not include experiments on WebArena, therefore we try our best to implement our own version and run on the same training tasks as our method.

### B.3.3    Implementation details of AUTOGUIDE

We directly run ReAct on the tasks with $\tau_+$ to collect $\tau_-$ actions and generate guidelines correspondingly. We provide a 5-shot prompt for the context identification module, which is shown in Figure 7. At test time, the top 2 guidelines at each timestep are selected to guide action generation.

| Parameter name | Value |
| --- | --- |
| Allowed Episode Length | 20 |
| n-shots | 2 |
| Agent Model | gpt-4-1106-preview |
| Context Identification Model | gpt-4-1106-preview |
| Guideline Selection Model | gpt-4-1106-preview |
| Guideline Extraction Model | gpt-4-1106-preview |
| top-k guideline selection | 2 |

Table 9: Experiment hyperparameters on WebArena. The number of shots follows the same setup in ReAct.

## B.4 Real Websites

### B.4.1 Environment details

We design a set of real-world website navigation tasks from 3 domains: Software Development (GitHub), Travel (Google Flights), and Education (Coursera), which have 30, 20, and 20 test tasks accordingly. Here are some example tasks:

- GitHub
  - Navigate to the repository for the Python library Seaborn with the most stars and show me all the open issues labeled with bug.
  - Go to the GitHub org for Spotify and open the pinned project with the most stars for me.
- Google Flights
  - Find me the one-way flight from Hong Kong to Paris departing on Oct 15th 2024 with the least emmisions.
  - Show me the booking options of the one-way flight departing from Auckland on September 11, 2024, and arriving in Rome with the earliest departure time on that day.
- Coursera
  - Show me a Cybersecurity course that can finish within 1 month and show me all the reviews for the selected course.
  - Find me a Coursera guided project that covers Unity and show me its main page.

We follow the action space design of Visual WebArena [5], which has the following action types available:

- goto [url]
- click [element_id]
- hover [element_id]
- type [element_id] [text] [1 for enter or 0 for not enter]
- press [key_combination]
- scroll [up or down]
- tab_focus [tab_index]
- close_tab
- go_back
- go_forward

As the real websites are constantly and dynamically changing, we evaluate the completed task with human experts.

### B.4.2 Baseline and Models

We directly run the two-shot SoM algorithm in the official codebase of Visual WebArena. The only modification we made from the original codebase is the bounding box detection algorithm, in

which we further filter invisible bounding boxes from the list and add the list elements as interactable elements to consider.

### B.4.3  Implementation details of AUTOGUIDE

We provide a total of 6 training tasks (3 for GitHub, 2 for Google Travel, and 1 for Coursera), on which we collect human demonstration as $\tau_+$'s and run SoM to collect $\tau_-$'s. From the pairs with multi-modal observations we generate text-based guidelines to guide action selection. Both context identification and guideline extraction are done by gpt-4-vision-preview, and we provide a 3-shot prompt for context identification. All parameter details are shown in Table 10.

| Parameter name | Value |
|---|---|
| Allowed Episode Length | 15 |
| n-shots | 2 |
| Agent Model | gpt-4-vision-preview |
| Context Identification Model | gpt-4-vision-preview |
| Guideline Selection Model | gpt-4-vision-preview |
| Guideline Extraction Model | gpt-4-vision-preview |
| top-k guideline selection | 2 |

Table 10: Experiment hyperparameters on multi-modal real-world website tasks.

## C  Prompt Templates

### C.1  Context Identification

We present our prompt templates for the context identification module $\mathcal{M}_{\text{context}}$ (Equation (1)) for WebShop, ALFWorld, and WebArena in Figures 5 to 7, respectively. In ALFWorld, there exist six categories of tasks, and we use context identification prompting with 2-shot examples for each task, following the practice by [23]. Figure 6 shows one example for the pick_and_place tasks.

### C.2  Guideline Extraction

Figures 8 to 10 detail our prompt templates $\mathcal{M}_{\text{guideline}}$ for extracting guidelines (Equation (2)) in the WebShop, ALFWorld, and WebArena domains.

### C.3  Context Matching

Figure 11 shows our prompt template for matching the generated context with one of the existing contexts if there is any similar context, for the construction of the set of context-aware guidelines (Section 3.2) and the retrieval of relevant guidelines during testing (Section 3.3) in all the three domains: WebShop, ALFWorld, and WebArena.

### C.4  Guideline Selection

For selecting only $k$ most relevant guidelines in case there are more corresponding context-aware guidelines during testing (Equation (3)), we use the prompt with Figure 12 for WebShop and ALFWorld and Figure 13 for WebArena.

## D  Example Context-Aware Guidelines

In Figures 14 and 15, we show a list of possible contexts and context-aware guidelines on WebArena and real-world websites, respectively.

**Context Identification Prompt for WebShop**

**Instruction**

Now, you'll get a snippet of a trajectory. Your job is to generate a brief and general summarization of the current status. Keep it broad and general, avoid any information about specific instructions, products, or buying options. For different 'Instructions' with the same status, the summarization should be the same.

**In-Context Example**

Webshop
Instruction:
i would like a bottle of bright citrus deodorant under 50 dollars
[Search]
SUMMARIZATION: The assistant is on the search page with a [Search] box.

Action: search[bright citrus deodorant]
Observation:
[Back to Search]
Page 1 (Total results: 3)
[Next $>$]
[B078GWRC1J]
Bright Citrus Deodorant
$10.99
[B078GTKVXY]
Ginger Fresh Deodorant
$10.99
[B08KBVJ4XN]
Barrel and Oak - Aluminum-Free Deodorant
$15.95
SUMMARIZATION: The assistant is on the search result page with a list of products.

Action: click[B078GWRC1J]
Observation:
[Back to Search]
[$>$ Prev]
scent [assorted scents][bright citrus][ginger fresh]
size [travel set (4-pack)][(pack of 1)][3-ounce (2-pack)]
Bright Citrus Deodorant
Price: $10.99
Rating: N.A.
[Description]

[Features]
[Reviews]
[Attributes]
[Buy Now]
SUMMARIZATION: The assistant is on a product's main page.

Action: click['bright citrus' '(pack of 1)']
Observation: Invalid Action!
SUMMARIZATION: The assistant failed to click on a button and got 'Invalid Action!'.

Action: click[bright citrus]
Observation: You have clicked bright citrus.
SUMMARIZATION: The assistant has successfully clicked on a button.

Action: click[Description]
Observation:
[Back to Search]
[$<$ Prev]
Bright Citrus Deodorant for sensitive skin
SUMMARIZATION: The assistant clicked on and navigated to a product's subpage(one of [Description], [Features], [Reviews], and [Attributes]).

Action: click[Buy Now]
Observation: Invalid Action!
SUMMARIZATION: The assistant failed to click on a button and got 'Invalid Action!'.

Action: search[bright citrus deodorant under 50]
Observation: Invalid Action!
SUMMARIZATION: The assistant failed to search on the page and got 'Invalid Action!'.

**Input**

Now it's your turn:
**{Current trajectory}**

Figure 5: Our prompt template for context identification (Equation (1)) in the WebShop domain.

**Context Identification Prompt for ALFWorld**

**Instruction**

You'll get a snippet of a trajectory of an text-based ALFRED task. Your job is to generate a brief and general summarization of the current status following 'SUMMARIZATION: '.
Keep it broad and general, avoid any information about specific objects and locations.
Here are two examples:

**In-Context Example**

You are in the middle of a room. Looking quickly around you, you see a cabinet 4, a cabinet 3, a cabinet 2, a cabinet 1, a coffeemachine 1, a countertop 1, a diningtable 3, a diningtable 2, a diningtable 1, a drawer 1, a fridge 1, a garbagecan 1, a microwave 1, a sidetable 1, a sinkbasin 1, and a toaster 1.
Your task is to: find some apple and put it in sidetable.
> think: To solve the task, I need to find and take an apple, then put it in sidetable.
OK.
> think: First I need to find an apple. An apple is more likely to appear in fridges (1), diningtables (1-3), sidetables (1), countertops (1), sinkbasins (1), garbagecan (1). I can check one by one, starting with fridge 1.
OK.
> go to fridge 1
The fridge 1 is closed.
SUMMARIZATION: Looking for an object here, but it is closed.
> open fridge 1
You open the fridge 1. The fridge 1 is open. In it, you see a lettuce 2, a mug 2, a potato 2, and a tomato 1.
SUMMARIZATION: Looking for an object, but it's not here.
> go to table 1
Nothing happens.
SUMMARIZATION: Tried to go to a location but failed.

> go to diningtable 1
On the diningtable 1, you see nothing.
SUMMARIZATION: Looking for an object, but it's not here.
> go to garbagecan 1
On the garbagecan 1, you see a apple 3, and a egg 3.
SUMMARIZATION: Looking for an object and found the desired object here.
> think: Now I find an apple (3). Next, I need to take it.
> take apple 3 from garbagecan 1
You pick up the apple 3 from the garbagecan 1.
SUMMARIZATION: Successfully took an object.
> think: Now I find an apple (3). Next, I need to put it in/on sidetable 1.
OK.
> go to sidetable 1
On the sidetable 1, you see nothing.
SUMMARIZATION: Looking for a location to put an object in/on.
> put apple 3 in/on sidetable
Nothing happens.
SUMMARIZATION: Tried to put an object in/on a location but failed.
> put apple 3 in/on sidetable 1
You put the apple 3 in/on the sidetable 1.
SUMMARIZATION: Successfully put an object in/on a location.

**In-Context Example**

…

**Input**

Now it's your turn:
**{Current trajectory}**

Figure 6: Our prompt template for context identification (Equation (1)) in the ALFWorld domain.

**Context Identification Prompt for WebArena**

**Instruction**

You are an autonomous intelligent agent tasked with navigating a web browser. You will be provided with the following information:
1. A list of context summarizations you have seen in the past.
2. A snippet of the current web page's accessibility tree: a simplified representation of the webpage with key information and the current web pages' URL.

Please generate the summarization of the current observation after 'SUMMARIZATION:'.

Here are some requirements:

Requirement 1: Different types of webpages should have clearly different summarizations. For example, for GitHub there can be the main page of GitHub, the overview page of a GitHub user, the issues page of a GitHub repository, the search result page of GitHub. etc, you should clearly categorize those and make sure not to mix them up.

Requirement 2: Important: The summarization should be general, and concise, without any user/object/task specific information, instead, websites that fall into the same categories should have the same summarization, for example, the main page of every reddit forum should be categorized as the same context summarization: On the main page of a Reddit forum. You should never include the specific name of the forum in the summarization.

Requirement 3: The URLs will be very useful for you to determine the summarization.

Requirement 4: If the context is the same as one from the seen list, directly copy the best matching one word by word.

**In-Context Example**

Observation:
The current web page's accessibility tree:
Tab 0 (current): Postmill
[1] RootWebArea 'Postmill' focused: True
    [31] HeaderAsNonLandmark ''
        [32] link 'Home'
    [55] link 'Forums'
    [56] link 'Wiki'
    [64] searchbox 'Search query'
    [65] link 'Notifications (0)'
    [66] link 'Submit'
    [12] button 'MarvelsGrantMan136' focused: True hasPopup: menu expanded: True
    [243] link ' Profile'
    [239] link ' My account'
    [245] link ' User settings'
    [255] link ' Block list'
    [286] separator '' orientation: horizontal
    [270] button ' Dark mode'
URL: http://reddit.com/
SUMMARIZATION: On the Reddit main page.

**In-Context Example**

...

**Input**

Here are the inputs:
**{Seen list}**
**{Current trajectory}**

Figure 7: Our prompt template for context identification (Equation (1)) in the WebArena domain.

**Guideline Extraction Prompt for WebShop**

**Instruction**

**{Task description}**. You will be provided with a desired and undesired trajectory of the same task. What is the first action that differs between the two trajectories? Why do you think it makes one trajectory failed and the other successful? Based on your answer, generate an action guideline to make future task avoid the same mistake. The guideline should specify what to do in what situation in the format of "When in what status, you should (or should not)...". On a product's page with product information, strictly refer to the option buttons as 'buying options such as sizes, colors, scents, and flavors', and clearly say that buying options are not subpages like [Description] and [Attributes] when you mention buying options. Your guideline must be general enough for any task, therefore never include any task-specific information, instead, refer to all the requirements as the requierments in Instruction. Strictly follow what the desired trajectory does and never suggest actions that the desired trajectory didn't do. When referring to actions, use the allowed action format. You should make your answer concise, limit your answer within 256 tokens, and put your answer in this format: 'Reasoning: ...
 Guideline: ...'.

**Input**

Desired Trajectory:
**{Desired trajectory}**
Undesired Trajectory:
**{Undesired trajectory}**

Figure 8: Our prompt template for guideline extraction (Equation (2)) in the WebShop domain.

**Guideline Extraction Prompt for ALFWorld**

**Instruction**

{Task description}. You will be provided with a desired and undesired trajectory of the same task. What is the first action that differs between the two trajectories? Why do you think it makes one trajectory failed and the other successful? Based on your answer, generate an action guideline to make future task avoid the same mistake. The guideline should specify what to do in what situation in the format of "When in what status, (optional: if you want to ...) you should (or should not)... (optional: a short example for demonstration. )". For the 'When in what status' part, directly use the words in SUMMARIZATION.
Here are two examples:
Example 1: When looking for an object, if you want to find a kitchen-related object like a spatula, you should start from the most possible locations.
Example 2: When looking for an object and found the desired object at the location, You should only take the exact object that you want.
Strictly follow what the desired trajectory does and never suggest actions that the desired trajectory didn't do. When referring to actions, use the allowed action format. You should make your answer concise, limit your answer within 128 tokens, and put your answer in this format: 'Reasoning: ... Guideline: ...'.

**Input**

Desired Trajectory:
{Desired trajectory}
Undesired Trajectory:
{Undesired trajectory}

Figure 9: Our prompt template for guideline extraction (Equation (2)) in the ALFWorld domain.

**Guideline Extraction Prompt for WebArena**

**Instruction**

{Task description}
You just finished a task but failed. For this failed task, we provide a human demonstration for you. Please compare the demonstration with your generated action at each step, reason about the intention of the correct action, and then geneate an action guideline for future tasks to avoid the same mistake and make the future tasks successful.

Here's the information you'll have:
1. The current observation:
* The task objective: the task you're trying to complete.
* The current web page's accessibility tree: a simplified representation of the webpage, providing key information.
* The current web page's URL: the link of the page you're currently on.
* The open tabs: the tabs you have opened.
* The previous actions: a sequence of past actions that you performed.

2. The action you generated in the failed run.

3. The correct action that you should take.

4. Demonstration actions in later steps on the same page.

Based on the information, please generate a short and concise guideline that guide you to issue the correct action.
Important: The guideline should be general enough to generalize to all similar tasks, not only this task. Therefore do not include any task-specific information in your guideline, for example a user name, a specific forum, the specific text you want to enter, or any number ID in [] in front of each element, for example [123], the numbers are randomly generated therefore never include them in your guideline. However, for other non-specific elements like

"link 'Forums'" or "button 'Create submission'", you should specifically include them in the exact text in your guideline.
When referring to a url in your guideline, specify it as detailed as possible, only replace the task specific information as a palceholder, for example, replace a forum iphone with <forum_name> and specify the url in full, starts with http://.
The guideline should be less than 128 tokens.
Please refer to "the previous actions" and "Demonstration actions in later steps" to generate more accurate descriptions of your purpose and the sequence of actions to achieve the purpose. make sure to emphasize the order of the actions, do not miss any single action, and put them in 1. 2. 3. ..., for example 'after you typed in all the text, you should do these sequentially: 1. ..., 2. ... . You must strictly follow the order."
When you mention multiple steps of actions, also mention in the guideline that you should refer to the PREVIOUS ACTIONS to reason about which actions you did and what you should do next. Specify that you should not repeatedly issue the same action, but should move on to the next action instead.
Only speicify what to do or what not to do, don't explain why.
It is important to clearly specify when to issue a stop action when the stop action is either the correct action or in the 'Demonstration actions in later steps on the same page.', do not specify the 'answer' in 'stop [answer]' because answer is different for different tasks, and do not mention anything about stop if this action is neither in "The correct action that you should take." nor "Demonstration actions in later steps on the same page.".
Please strictly adhere to the 'correct action that you should take', do not propose other actions.

**Input**

Here are the information you need:
{Observation}
The action you generated in the failed run:
{Predicted action}
The correct action that you should take:
{Demo action}
Demonstration actions in later steps on the same page:
{Later action}
Please put your answer in this format: Reasoning: ... Guideline: ...

Figure 10: Our prompt template for guideline extraction (Equation (2)) in the WebArena domain.

**Context Matching Prompt**

**Instruction**

{Task description}
A task trajectory can be long. Therefore the assistant summarizes the status of each step.
For different task with the same status, the summarization should be the same, therefore please ignore any information about instructions or products.
You will be provided with the following:
1. A list of summarizations the assistant saw in the past.
2. A newly generated summarization.
Please determine if any summarization from the list matches the exact same status as the newly generated one. If yes, answer the index of the corresponding summarization, for example "Answer: 2"; otherwise, "Answer: None".

**Input**

Seen Summarizations:
{List of contexts}

Figure 11: Our prompt template for context matching (Sections 3.2 and 3.3) in all the WebShop, ALFWorld, and WebArena domains.

**Guideline Selection Prompt for WebShop and ALFWorld**

**Instruction**

{Task description}. You will be equipped with the following resources:
1. A list of action guidelines with valuable guidelines.
2. Trajectory history, which includes recent observations and actions.
Not all guidelines are useful to generate the next action. Please select the guidelines that are useful and relevant to the next action given the trajectory and recent observations. To generate the next action, which guidelines from the provided guidelines are most useful to directly tell you what to do for the next action? You can select up to 2 guidelines, and put the indices of the selected guidelines in a python list. For example if you select guideline 1, 5, answer: [1, 5]. If none of them are useful for generating the next action, answer the empty list [].

**Input**

{List of guidelines}
{Current trajectory}

Figure 12: Our prompt template for selecting the most relevant context-aware guidelines during the test time (Equation (3) from Section 3.3) in the WebShop and ALFWorld domains.

**Guideline Selection Prompt for WebArena**

**Instruction**

{Task description}. At each time step, you need to generate one action given the current observation.
You will be equipped with the following resources:
1. A list of action guidelines with valuable guidelines.
2. The intent of the task, which is the objective/goal that you should achieve.
3. Trajectory history, which includes the current observation and a sequence of past actions.
Not all guidelines are useful to generate the next action. Please select the guidelines that are useful and relevant to the next action given the current observation and past actions.
To generate the next action, which guidelines from the provided guidelines are most useful to directly tell you what to do for the next action? You can select 3 guidelines (or less if there are less than 3 guidelines), and put the number indices of the selected guidelines in a python list. For example if you want to select guideline 2 and 5, answer [2, 5]. If none of them are relevant, answer [].

**Input**

{List of guidelines}
{Current trajectory}

Figure 13: Our prompt template for selecting the most relevant context-aware guidelines during the test time (Equation (3)) in the WebArena domain.

**Context:** On the Reddit main page.
**Context-Aware Guideline:**
- When on the Reddit main page, if you want to change your bio, you should click on the "link 'Profile'", which is located in the user menu dropdown, right above the "link 'My account'". The correct action format to do this is ```click [profile_link_id]```.
- When on the Reddit main page, if you want to navigate to a specific forum, you should click on the "link 'Forums'", which is located at the early part of the observation, right above the link 'Wiki'. The correct action format to do this is ```click [link_id]```.
- When on the Reddit main page, if you want to create a new forum, you should click on the "link 'Forums'", which is located near the top of the observation, right above the "link 'Wiki'". The correct action format to do this is ```click [link_id]```.
- When on the Reddit main page, if you want to like all submissions created by a specific user in a specific subreddit, you can directly navigate to the user's page with the action format ```goto [url]```, replacing [url] with the user's page URL, formatted as http://<platform_domain>/user/<username>.

**Context:** On the overview page of a Reddit user.
**Context-Aware Guideline:**
- When on the overview page of a Reddit user, if you want to interact with submissions from a specific subreddit, you should first navigate to the 'Submissions' tab to filter the content by the user's submissions. The correct action format to do this is ```click [link_id]```, where [link_id] is the ID of the 'Submissions' link in the main content area of the page.

**Context:** On the biography edit page of a Reddit user.
**Context-Aware Guideline:**
- When on the biography edit page of a Reddit user, if you want to change the biography text, you should do these sequentially: 1. Click on the textbox 'Biography' to focus it, located in the main section of the page, with action ```click [textbox_id]```. 2. Select all text inside the textbox using the action ```press [Meta+a]```. 3. Clear the selected text with the action ```press [Backspace]```. 4. Type the new biography content into the textbox 'Biography' with action ```type [textbox_id] [new_content] [1]```. 5. Click on the button 'Save', located below the textbox 'Biography', to submit the changes with action ```click [button_id]```. After these steps, issue a stop action when the task is complete.

**Context:** On the page listing all forums on a Reddit-like platform.
**Context-Aware Guideline:**
- When on the page listing all forums on a Reddit-like platform, if you want to navigate to a specific forum, you should do these sequentially: 1. click on the "link 'Alphabetical'", which is located in the main area, to sort forums alphabetically. The correct action format to do this is ```click [link_id]```. 2. After the forums are sorted, click on the specific "link '<forum_name>'" that you wish to navigate to. The correct action format to do this is ```click [link_id]```.

**Context:** On the Reddit page to create submission.
**Context-Aware Guideline:**
- When on the Reddit page to create submission, if you have filled in the 'Title' and 'Body' textboxes but do not see the button 'Create submission', you should scroll down to reveal more of the page. The correct action format to do this is ```scroll [down]```. After scrolling, if you want to submit the post, you should click on the button 'Create submission', which is located after the 'Body' textbox. The correct action format to submit the post is ```click [button_id]```.

**Context:** On the main page of a Reddit forum.
**Context-Aware Guideline:**
- When on the main page of a Reddit forum, if you want to find posts related to a specific topic among the top posts, you should first sort the posts by their popularity to ensure you are viewing the most relevant content. To do this, you can: 1. click on the "button 'Sort by: Hot'", which is located in the main section of the page, below the forum heading and above the first article. The correct action format to do this is ```click [id]```. 2. After the sorting options have expanded, click on the "link 'Top'", which will appear as a new option under the sorting button. This action should be repeated once. The correct action format to do this is ```click [id]```.
- When on the main page of a Reddit forum, if you want to create a new post, you should click on the "link 'Submit'", which is located in the early part of the observation, right below the "link 'Notifications (0)'". The correct action format to do this is ```click [link_id]```.

**Context:** On the submissions page of a Reddit user.
**Context-Aware Guideline:**
- When on the submissions page of a Reddit user, if you want to perform an action on specific subreddit submissions but do not see them, you should scroll down to reveal more submissions. The correct action format to do this is ```scroll [down]```. After scrolling, if you find a submission from the desired subreddit, such as 'UpliftingNews', and need to downvote it, click on the button 'Downvote' located at the end of the submission's details. The correct action format to downvote is ```click [downvote_button_id]```. Once all required actions are performed on the submissions, issue a stop action with the format ```stop []```.

**Context:** On the submissions page of a Reddit forum sorted by top.
**Context-Aware Guideline:**
- When on the submissions page of a Reddit forum sorted by top, if you want to review posts but see "There's nothing here…", you should expand the time range to view more posts. Do this by clicking on the button 'From: Past 24 hours', which is located in the main section of the page, below the heading '/f/books' and above the StaticText "There's nothing here…". The correct action format to do this is ```click [button_id]```. After expanding the time range, if you find a post that meets the task criteria, issue a stop action.

**Context:** On the submissions page of a Reddit forum sorted by new.
**Context-Aware Guideline:**
- When on the submissions page of a Reddit forum sorted by new, if you want to upvote the newest post and you have already clicked the upvote button for the first article entry, you should issue the stop action. The correct action format to do this is ```stop```.

**Context:** On the page of a Reddit post.
**Context -Aware Guideline:**
- When on the page of a Reddit post, if you have already navigated to the 'Submit' link, filled in the image URL and title, and clicked the 'Create submission' button, you should consider the task complete. The correct action format to do this is ```stop```.

Figure 14: Example contexts and corresponding guidelines for WebArena.

**Context:** On the main page of GitHub.
**Context-Aware Guideline:**
- When on the main page of GitHub, if you want to search for a repository, you should type the search query into the search bar at the top of the page, which is visually identifiable and typically labeled with text like 'Search or jump to...'. Do not type into any other input fields. Perform the action as follows: ```type [search bar id] [search query] [1]```.

**Context:** On the issues page of a GitHub repository.
**Context-Aware Guideline:**
- On the issues page of a GitHub repository, if you want to filter issues by a specific label, you can type the label filter in the search input field, which is usually at the top of the current webpage and shown as the first appeared "[INPUT] []" in observation. The correct action format to do this is ```type [id] [label:"specific_label"] [1]```. After applying the filter: 1. Click on the link that shows the number of closed issues, which is labeled as "[A] [num Closed]" in observation, located near the search input field. The correct action format is ```click [link_id]```.

**Context:** On the search result page of GitHub.
**Context-Aware Guideline:**
- On the search result page of GitHub, if you want to filter the search results to find a specific organization, after you have typed the organization's name in the search bar, you can do these sequentially: 1. click on the "[LI] [Users]" in the left sidebar, which is visually represented by the blue text "Users" and is the first "[LI] [Users]" in observation, by action ```click [li_id]``` 2. if the user or organization filter is applied, click on the organization's name, which is visually represented by the blue text "Meta" under the "Users" section, by action ```click [link_id]```

**Context:** On the search result page of Coursera.
**Context-Aware Guideline:**
- On the search result page of Coursera, if you want to select a course, after you have typed the course topic in the search bar, you can do these sequentially: 1. click on the filter for courses, which is represented by "[INPUT] []" and visually located in the filter section on the left side of the webpage, by action ```click [input_id]``` 2. click on the course link, which is represented by "[A] [course_name]" and visually located in the main content area of the webpage, by action ```click [link_id]```. This action should be repeated twice. 3. switch to the new tab that contains the course details by ```page_focus [1]```.

**Context:** On the main search page of Google Flights.
**Context-Aware Guideline:**
- On the main search page of Google Flights, if you want to search for a one-way flight with specific departure and destination airports and date, you can do these sequentially: 1. type the departure airport code in "[INPUT] [Where from?]", which is the first input box at the top of the search area, by action ```type [element_id] [airport_code] [1]``` 2. type the destination airport code in "[INPUT] [Where to?]", which is next to "[INPUT] [Where from?]", by action ```type [element_id] [destination_airport_code] [1]``` 3. click on "[DIV] [Round trip]" to change the trip type, located at the top of the search area, by action ```click [element_id]``` 4. click on "[LI] [One way]" to select the one-way trip option, which appears after clicking "[DIV] [Round trip]", by action ```click [element_id]``` 5. type the departure date in "[INPUT] [Departure]", which is next to "[INPUT] [Where to?]", by action ```type [element_id] [date] [1]``` 6. click on "[BUTTON] [Done]" to confirm the date, which appears after entering the departure date, by action ```click [element_id]``` 7. click on "[BUTTON] [Search]" to perform the flight search, which is below the search fields, by action ```click [element_id]```

**Context:** On the search result page of Google Flights with a list of Best departing flights and Other departing flights.
**Context-Aware Guideline:**
- On the search result page of Google Flights with a list of Best departing flights and Other departing flights, if you want to select a flight, you can click on the first flight option under the "Best flights" section. The correct action format to do this is ```click [flight_option_id]```, where [flight_option_id] is the id of the [LI] element corresponding to the first flight listed. This [LI] element is visually located at the top of the list of flights and contains the departure and arrival times, airline name, flight duration, and other details.
- On the search result page of Google Flights with a list of Best departing flights and Other departing flights, if you want to locate the flight with the least emissions, you can do these sequentially: 1. Click on the sort options button, which is visually located at the top of the flight list and shown as "[BUTTON] [Sort by:]" in observation, by action ```click [sort_button_id]```. 2. Then, click on the emissions sort option, which is visually located in the sort options dropdown and shown as "[LI] [Emissions]" in observation, by action ```click [emissions_option_id]```. 3. Finally, click on the first flight listed under the "Best flights" section, which is visually located at the top of the list and shown as "[LI] [flight_details]" in observation, by action ```click [first_best_flight_id]```. Repeat this action twice.

Figure 15: Example contexts and corresponding guidelines for real-world websites.

