# OpenReview forum: "AutoGuide: Automated Generation and Selection of Context-Aware Guidelines for Large Language Model Agents"
_NeurIPS.cc/2024/Conference — NeurIPS 2024 poster_

### Official Review · Reviewer_wFet · 2024-07-12

**Soundness:** 3
**Presentation:** 3
**Contribution:** 3
**Rating:** 7
**Confidence:** 4

**Summary:**

This paper proposes a decision making framework called AudoGuide. It leverages domain knowledge from offline experiences to generate context-aware guidelines. This framework improves LLM agents in downstream decision-making tasks.

**Strengths:**

1.Design of context aware guidelines is ingenious.
2. The proposed framework outperforms competitive baselines in complex benchmarks.

**Weaknesses:**

1. The two advantages when comparing with related works seems contradictory, e,g., inter-task knowledge compared with Reflexion (line 69), context-aware guideline compared with ExpeL (line 83). As the context-aware guideline and inter-task knowledge both contribute to the performance, does it imply that there're sharing contexts across different tasks? Then what's the percentage of inter-task sharing CONTEXT, and inter-task guideline retrieved by guideline selection?

2. Analysis in section 4.2.Q1 (line 222-227) is inconsistent with Table 1. It seems to analyze "ReAct+AutoGuide" over "ReAct+ExpeL" by claming "ExpeL approach helps ReAct by ... but is not as significant as AUTOGUIDE". But I can't find "ReAct+ExpeL" setting in Table 1. It's confusing since following discussion also centers around this setting.

3. Lack of explanation about the setting "AUTOGUIDE + Reflexion" in section 4. It's simply mentioned in section 4.1.2 (line 197). How is it implemented?

4. "Context Identification" module necessitates handcrafted context definitions per task. Since guideline selection relies on context matching, will the number of demonstrations affect task performance?

**Questions:**

See above.

A typo: line 110 Appendix C.1: -> C.1.

**Limitations:**

See above.

---

> ### Author Rebuttal · Authors · 2024-08-07
>
> We appreciate the reviewer for the positive evaluation of our paper and providing constructive comments. We have addressed individual comments below and conducted additional evaluations. We will also carefully incorporate your feedback into an updated version of our paper.
>
> > “... As the context-aware guideline and inter-task knowledge both contribute to the performance, does it imply that there're sharing contexts across different tasks? Then what's the percentage of inter-task sharing CONTEXT, and inter-task guideline retrieved by guideline selection?”
>
> We hope to clarify that both contexts and guidelines are shared across different tasks. Below, for each context, we count the number of tasks where the same context is shared and then calculate the average percentage in ALFWorld. We apply the same calculation for guidelines:
>
> | | Average Shared Percentage (%) |
> | :---------------- | :----: |
> | Context | 48.82% |
> | Guideline | 43.64% |
>
> > “... But I can't find "ReAct+ExpeL" setting in Table 1 …”
>
> Thank you for your feedback. The "ExpeL" baseline in Table 1 corresponds to the "ReAct + ExpeL" baseline. We will correct this confusion by changing the "ExpeL" baseline in Table 1 to "ReAct + ExpeL" in the paper.
>
> > “Lack of explanation about the setting "AUTOGUIDE + Reflexion" in section 4”
>
> We appreciate your feedback. We will provide more detailed information about our implementation of Reflexion, including the maximum number of trials (3), the GPT version (GPT-4-Turbo), the temperature setting (0.0), and the number of shots (2 for ALFWorld and 0 for WebShop), in Section 4.
>
> > "Context Identification" module necessitates handcrafted context definitions per task. Since guideline selection relies on context matching, will the number of demonstrations affect task performance?
>
> In response to your insightful comment, we introduce a new experiment that eliminates the use of handcrafted context definitions in our context identification module. Specifically, we devise a general prompt (please refer to below “Context Proposal Prompt for WebArena”), where GPT-4-Turbo proposes a possible set of context summaries. Then, we use GPT to replace the human-designed list of context summaries (e.g., “On the main page of reddit” in Figure 7) with the ones proposed by GPT (e.g., “On a forum main page with navigation links and user menu”; please refer to below “GPT Proposed Contexts for WebArena”). We show that employing AutoGuide with GPT-proposed contexts in the few-shot examples (rather than human-designed contexts as in Figure 7) achieves comparable results to the original AutoGuide on WebArena:
> | Algorithm | Success Rate (%) |
> | :---------------- | :----: |
> | AutoGuide – Human-Designed Contexts | 47.1% |
> | AutoGuide – GPT-Proposed Contexts | 46.0% |
>
> We additionally conduct a new experiment on WebArena, varying the number of demonstrations in the human-designed context identification prompt. The result shows that our context identification module is robust to the number of demonstrations:
> | Number of Demonstrations ($n$) | Success Rate (%) |
> | :---------------- | :----: |
> | $n=1$ | 44.9% |
> | $n=3$ | 47.1% |
> | $n=5$ (Original Hyperparameter) | 47.1% |
>
> > ‘A typo: line 110 Appendix C.1: -> C.1.”
>
> Thank you for noting this. We will fix this typo in the paper.
>
> ---
> ### Context Proposal Prompt for WebArena
> ```
> (Instruction) You are an autonomous intelligent agent for general-purpose text-based interactive tasks. You will be provided with a few trajectories of tasks from an interactive environment, each trajectory includes a goal, and a sequence of observations and actions. For each observation, we want you to abstract the underlying context at that step into a short, concise, and general context summarization, which describes the context given the goal and target.
> For example:
> 1. On the main page of Google Scholar with a [search] box.
> 2. Attempted to chop an object but failed, therefore getting an error message as observation.
> Notice: Your summarization must be general enough, so that similar situations, especially similar ones in consecutive steps, get summarized into the same context identification. Therefore, please do not include any task-specific information, for example, do not mention a specific user name, specific product category, or specific object type, instead conclude them to a broad and general category: a user, a product or an object. Please refer to attempted actions in the past tense and use the same words for observation and actions in the trajectories. Please list all the unique context summarizations in a python list: Answer: ['context_summarization1', 'context_summarization2', 'context_summarization3', ...]. Please don't put a context_summarization that doesn't appear in the provided trajectories.
>
> (Trajectory Format Specification) The goal is after "Intent:", observations are after "Observation: ", including an accessibility tree of a focused webpage and its corresponding URL, and actions are after "Action: ".
>
> (Input) Here is the input: {Trajectories}
> ```
>
> ### GPT Proposed Contexts for WebArena
> ```
> On a forum main page with navigation links and user menu
> On a forum list page with navigation links and sorting options
> On a forum category page with navigation links and submission options
> On a submission preview page with options to edit or submit
> On an alphabetical list of forums with navigation options
> ```

---

> > ### Comment · Reviewer_wFet · 2024-08-09
> >
> > Thank you for your thorough explanation and experiments! My concerns towards weakness 1/2/4 are all addressed.
> >
> > Regarding the setting "AUTOGUIDE + Reflexion", my question is that how to combine/integrate them together, since they're different frameworks with multiple individual components.

---

> > > ### Author Response · Authors · 2024-08-10
> > > **Response to Reviewer wFet**
> > >
> > > We are pleased that our rebuttal addressed concerns #1, #2, and #4. We also appreciate the reviewer's clarification of concern #3. Please find further details on the AutoGuide + Reflexion approach below.
> > >
> > > In the initial episode of a given test task, we begin by using AutoGuide to solve the task. If the initial episode fails, we then use Reflexion to generate reflective feedback based on the test reward. This feedback suggests the plan the agent should attempt in the next episode (or trial). In our AutoGuide + Reflexion setting, **this reflective feedback is appended to the action generation prompt, immediately following the context-aware guidelines**. This approach enables the agent to consider both the context-aware guidelines and the reflective feedback when solving the test task in the next episode. As in Reflexion, when there are multiple past episodes, we concatenate the reflection feedback from each episode. This process continues until the maximum number of episodes (or trials) is reached.

---

> > > > ### Comment · Reviewer_wFet · 2024-08-10
> > > >
> > > > Thank you for the explanation!
> > > >
> > > > My concerns are addressed.  Please also provide the implementation details in the revised version.

---

> > > > > ### Author Response · Authors · 2024-08-12
> > > > > **Response to Reviewer wFet**
> > > > >
> > > > > We're glad our responses addressed your concerns. We also appreciate your positive evaluation of our paper. Yes, based on your insightful and helpful feedback, we will include the implementation details in the revised version.

---

### Official Review · Reviewer_rEoJ · 2024-07-12

**Soundness:** 3
**Presentation:** 3
**Contribution:** 3
**Rating:** 6
**Confidence:** 3

**Summary:**

The paper introduces AUTOGUIDE, a novel framework designed to enhance LLM agents' performance in unfamiliar domains like web navigation by automatically generating context-aware guidelines from offline experiences. These guidelines are expressed in concise natural language and follow a conditional structure, clearly describing the applicable context. The framework includes two key modules: the context identification module, which abstracts the agent's state into a concise description, and the guideline extraction module, which generates the desired guideline for a specific context. The evaluation demonstrates that AUTOGUIDE significantly outperforms competitive baselines in complex benchmark domains, including real-world web navigation.

**Strengths:**

1. The method proposed is reasonable. It acts like a kind of "library learning" that summarizes past experiences into reusable modules and improves the agents' ability in unfamiliar domains.
2. The experimental results are promising, showing the advantage of the proposed method.
3. The writing is clear.

**Weaknesses:**

1. The author claims that their method and the in-context example selection module in ExpeL are orthogonal, so they only compared ExpeL with guidelines. This point requires experimental verification.
2. The author should provide the number of tokens used or the number of times the LLM is called (in training and inference), to more comprehensively compare the various methods.

**Questions:**

1. How is the number of training tasks determined for different domains?

**Limitations:**

The authors have discussed the limitations well.

---

> ### Author Rebuttal · Authors · 2024-08-07
>
> Thank you for your constructive review and helpful questions. We have addressed each comment individually and conducted additional experiments based on your insightful feedback. We will also carefully incorporate your feedback into an updated version of our main paper and appendix.
>
> > “The author claims that their method and the in-context example selection module in ExpeL are orthogonal, so they only compared ExpeL with guidelines. This point requires experimental verification.”
>
> In response to your insightful feedback, we conduct two additional experiments on ALFWorld: AutoGuide with the In-Context Example Selection Module and ExpeL with the In-Context Example Selection Module. The results are summarized in the table below, leading to the following conclusions:
> - AutoGuide can be readily expanded with other techniques, such as in-context example selection.
> - AutoGuide consistently outperforms ExpeL, both with and without the in-context example selection module.
>
> | Algorithm | Success Rate (%) |
> | :---------------- | :----: |
> | ExpeL | 59.0% |
> | ExpeL + In-Context Example Selection | 61.9% |
> | AutoGuide | 79.1% |
> | AutoGuide + In-Context Example Selection | 85.8% |
>
> > “The author should provide the number of tokens used or the number of times the LLM is called (in training and inference), to more comprehensively compare the various methods.”
>
> Please refer to the tables below for the number of tokens used during training and inference. We compute these statistics using a subset of training and testing tasks on WebArena. For training, we observe that AutoGuide requires more tokens than ExpeL due to the additional context generation. However, please note that this context generation enables the filtering of irrelevant guidelines during inference. As a result, AutoGuide uses a comparable number of tokens to ExpeL (which provides all guidelines at inference to an LLM agent) and achieves higher test performance than ExpeL.
>
> | Algorithm | Average Token per Training Data |
> | :---------------- | :----: |
> | ReAct | N/A |
> | ExpeL | 3888.2 |
> | AutoGuide | 4873.4 |
>
> | Algorithm | Average Token per Inference Step |
> | :---------------- | :----: |
> | ReAct | 1734.6 |
> | ExpeL | 5125.5 |
> | AutoGuide | 4809.1 |
>
> > “How is the number of training tasks determined for different domains?”
>
> Generally, the more training tasks available, the better, as a larger offline dataset can enhance AutoGuide's performance. In practice, we determine the number of training tasks for each domain based on several factors: the number of available offline experiences/tasks and budget considerations.

---

> > ### Comment · Reviewer_rEoJ · 2024-08-10
> >
> > Thanks for the response! I've read it and will keep the score for acceptance.

---

> > > ### Author Response · Authors · 2024-08-12
> > > **Response to Reviewer rEoJ**
> > >
> > > Thank you for reading our rebuttal. We also appreciate your positive review and kind support of our paper.

---

### Official Review · Reviewer_zgYy · 2024-07-13

**Soundness:** 3
**Presentation:** 3
**Contribution:** 3
**Rating:** 6
**Confidence:** 4

**Summary:**

This paper introduces AUTOGUIDE, a framework for enhancing large language model agents' performance in sequential decision-making tasks by automatically generating context-aware guidelines from offline experiences. The method consists of two main components: a context identification module and a guideline extraction module. These modules work together to create concise, natural language guidelines that are relevant to specific contexts. At test time, AUTOGUIDE identifies the current context and selects pertinent guidelines to assist the LLM agent in decision-making. The authors evaluate AUTOGUIDE on various benchmark domains, including ALFWorld, WebShop, WebArena, and real-world multi-modal websites.

**Strengths:**

- Context-awareness is noteworthy because it addresses a fundamental challenge in leveraging offline experiences for LLM agents. By generating guidelines that are explicitly tied to specific contexts, AUTOGUIDE can provide more targeted and relevant assistance during decision-making. This is evident in the example shown in Figure 4, where AUTOGUIDE's context-aware guideline helps the agent locate a soapbar in a less obvious place (the toilet).
- Ablation study is pretty strong. The authors test their method across a diverse range of benchmark domains, including both text-based and multi-modal environments, demonstrating AUTOGUIDE's versatility and effectiveness.

**Weaknesses:**

- The method can be summarized as ReAct with context identification and guideline retrieval. And from the ablation study, it seems like the GES component is really important. For each different task, the guidelines have to be constructed, and that can introduce noise. This may hinder the effectiveness of the method.

**Questions:**

N/A

---

> ### Author Rebuttal · Authors · 2024-08-07
>
> We greatly appreciate your positive evaluations of our paper and insightful feedback. Below, we respond to your valuable comment. We will also carefully incorporate your feedback into an updated version of our main paper and appendix.
>
> > “For each different task, the guidelines have to be constructed, and that can introduce noise. This may hinder the effectiveness of the method.”
>
> First, we hope to clarify how AutoGuide generates context-aware guidelines across different domains and tasks:
> - For each domain (e.g., ALFWorld, WebArena, WebShop), a new set of contexts and guidelines is extracted.
> - Within a domain, contexts and guidelines are extracted from training tasks and then used for test tasks.
> - For each state of a test task, the context is identified and the most helpful guidelines are selected.
>
> AutoGuide can experience noise from three sources: 1) noise in the context identification module, 2) noise in guideline selection module, and 3) noise in guideline extraction module. In this rebuttal, we conduct an additional analysis to demonstrate that the noise in each module is minimal.
>
> For the noise sources of 1) and 2), we provide statistics on the rate at which the context identification module incorrectly identifies contexts and the guideline selection module selects incorrect guidelines on ALFWorld through manual evaluation. This analysis demonstrates that both the context identification and guideline selection modules produce results with low noise:
>
> | Module | Error Rate (%) |
> | :---------------- | :----: |
> | Context Identification | 3.2% |
> | Guideline Selection | 1.1% |
>
> For the noise source of 3), it is challenging to determine the correctness of the guideline extraction module compared to the first two modules (i.e., context identification and guideline selection). Therefore, we qualitatively examine each extracted guideline in Figures 14-15. Our observations indicate that these guidelines contain helpful and effective knowledge, as demonstrated in our empirical experiments. Having a standardized metric for quantifying the quality of generated guidelines would be beneficial, and this is part of our planned future work, as noted in Appendix A (Limitation and Broader Impacts).
>
> Lastly, AutoGuide employs principled methods to systematically address potential noise. Specifically, AutoGuide includes a context-matching procedure (Lines 128-131) to determine if the currently generated context matches any previously identified contexts. Additionally, AutoGuide uses the guideline selection module (Section 3.3) to choose up to $k$ relevant guidelines based on the current context. If no guidelines are deemed relevant, this guideline selection module can opt not to select any guideline.

---

### Decision · Program_Chairs · 2024-09-25

**Decision:**

Accept (poster)

**Comment:**

This paper proposes a method (autoguide) to improve LLM agent performance by generating context-aware guidelines.

Strengths:
S1. All reviewers like the idea of context-aware guideline.
S2. Strong experimental results on multiple benchmark tasks.

Weakness:
W1. Some connections and comparisons with ReAct, Reflexion, ExpeL need further evidence.

Overall, it is a clearly-written paper with an interesting idea (context-aware guideline).